# National-Scale Cobalt Geochemical Mapping of Exposed Crust in China

**Dongsheng Liu [1,2], Qinghua Chi [1,2], Xueqiu Wang [1,2], Yuanyuan Chen [3], Lanshi Nie [1,2,*] and Fan Yang [1,2,*]**

1   Key Laboratory of Geochemical Exploration, Institute of Geophysical and Geochemical Exploration (IGGE), Chinese Academy of Geological Sciences, Langfang 065000, China
2   International Centre on Global-Scale Geochemistry (ICGG), Langfang 065000, China
3   Hebei Regional Geological Survey, Langfang 065000, China
*   Correspondence: nielanshi@igge.cn (L.N.); yf51318@163.com (F.Y.)

**Abstract:** Geochemical background is the basis of anomaly evaluation in mineral exploration and environmental investigation. However, the background value obtained from secondary media is inevitably affected by weathering, transportation, and sorting, which leads to secondary depletion or enrichment of chemical elements. This problem can be avoided by the geochemical mapping of the exposed crust. In this paper, more than 38,000 samples of rocks data were collected, and an original method was presented for the first time to produce the cobalt geochemical map of exposed crust across the whole of China. Using a geological map grid of 1:200,000 as the basic calculation unit, the area and content of rock types in each unit were calculated, and then the grid was assigned using the area-weighted average method. Based on this, an geochemical contour map was drawn in ArcGIS. The results show that the median of the exposed crust is 9.74 mg/kg, which is lower than that of the whole crust and soils. This value is explained by the vertical heterogeneity of the crust and the secondary enrichment, respectively. Co anomalies and anomalous centers of exposed crust are distributed in southwest, north, and northwest China, where they are primarily related to Permian Emeishan Large Igneous Province (ELIP), Pacific plate, and the big mantle wedge (BMW), Archean metamorphic basic rocks, and ophiolite belts.

**Keywords:** petrogeochemical mapping; China; cobalt; abundance of exposed crust





## 1. Introduction

The background value of elements is the relentless pursuit of geochemical mapping scientists. The regional geological background value is a benchmark for evaluating the level of local-scale anomaly, so it is of great significance for geochemical exploration and environmental evaluation. However, theoretically, only an approximation of the background value can be obtained by determining the chemical element content in secondary media (soils, stream sediments, etc.) [1–3]. Two main factors prevent us from obtaining a true background value through secondary media. First, the weathering process of rocks inevitably leads to secondary enrichment or depletion of chemical elements, which leads to element content in secondary media deviating from the real geological background [4]. Although the variation in element content caused by the weathering process can be partially corrected by the empirical formula [5], the empirical formula itself also needs to be modified with the change in the surface environment. Second, mineralization or anthropic pollution causes a deviation in the background value. Although many kinds of data processing methods have been developed to identify and exclude the influence of mineralization or anthropic pollution, the deviation may not be eliminated entirely [6]. Collecting primary media (sedimentary rocks, igneous rocks, metamorphic rocks, etc.) to calculate background values rather than secondary media can avoid these problems. The regional upper crust abundance given by previous researchers is the overall element content of the region, which cannot reflect the spatial variation of the region. The change in the natural background of

chemical elements is so important that it is necessary to describe their background values, which may change with space [7]. Carrying out petrogeochemical mapping by collecting sufficiently representative rock samples can provide reliable information on the spatial variation of geochemical background.

In addition to establishing a geochemical background, petrogeochemical mapping can provide support for the study of global element distribution and circulation. The distribution and cycle of key elements is one of the 10 priority scientific issues in Earth science in the next decade listed by NASEM [8]. The transfer of chemical elements from rock to soil/stream sediment is an important process of element surficial circulation. In this process, element content and mineral composition in media are affected by various mechanisms, such as weathering, transport, sorting, and deposition [9,10]. Comparing the contents of elements between primary and secondary media on the regional scale will provide support for basic research on the distribution and cycle of elements. At present, a large number of geochemical maps of secondary media have been mastered, but petrogeochemical maps are lacking.

Due to the drastic spatial variation of rock composition, to solve the problem of sampling representativity, producing a petrogeochemical map needs the joint support of massive data on rock element composition and rock distribution information, which will cost a huge amount of money, time, and manpower. At present, only Chi et al. (2005) produced a petrogeochemical map of eastern China covering about 3.3 million km$^2$ [11]. No complete national-scale petrogeochemical map based on measured data has yet been produced in China or anywhere in the world.

To produce the petrogeochemical map, we collected data from more than 38,000 representative rock samples, as well as information on the spatial distribution of rocks nationwide. In this paper, among many elements, we chose cobalt as an example to produce national-scale petrogeochemical mapping. The reason for choosing cobalt is that its content varies greatly with lithology but is evenly distributed in the same lithology [12–14]; that is, the cobalt content of rocks has a good correspondence with rock types, which is conducive to solving representativity problems. Another reason for choosing this transition metal element as an example is that cobalt resources have received extensive attention in recent years because of the dramatic increase in demand [15], and they have been put on the list of critical materials or strategic minerals by many countries around the world [16,17]. The Democratic Republic of the Congo, South Africa, Australia, Philippines, etc. are major cobalt ores exporters [18], and China has been a leading consumer and refiner of cobalt [19]. For most countries, including China, the recycling efficiency of cobalt is relatively low, which means that large amounts of cobalt are released into the environment [18]. The pollution of cobalt as a heavy metal has also been a concern in recent years [20,21].

## 2. Methods

Data from 38,899 rock samples were used in this paper. Sample collection was completed in two stages: 1986–1996 and 2008–2012 (Figure 1). All rock samples were collected from the surface outcrops, so petrogeochemical mapping is the geochemical mapping of the exposed crust.

### 2.1. Sampling and Analytical Methods

2.1.1. Collection of Sedimentary Rock Samples

The collection of rock samples was deployed along the standard stratigraphic section based on each 1:200,000 geological map. When the standard stratigraphic section do not meet the requirements, partial auxiliary or multi-point sampling can also be arranged. The spatial distribution should be distributed as evenly as possible. The area of each type of stratigraphic unit in each 1:200,000 geological map was calculated. The collected samples should be able to represent the main stratigraphic units in each 1:200,000 geological map, covering more than 80% of the bedrock outcrop area of the map.

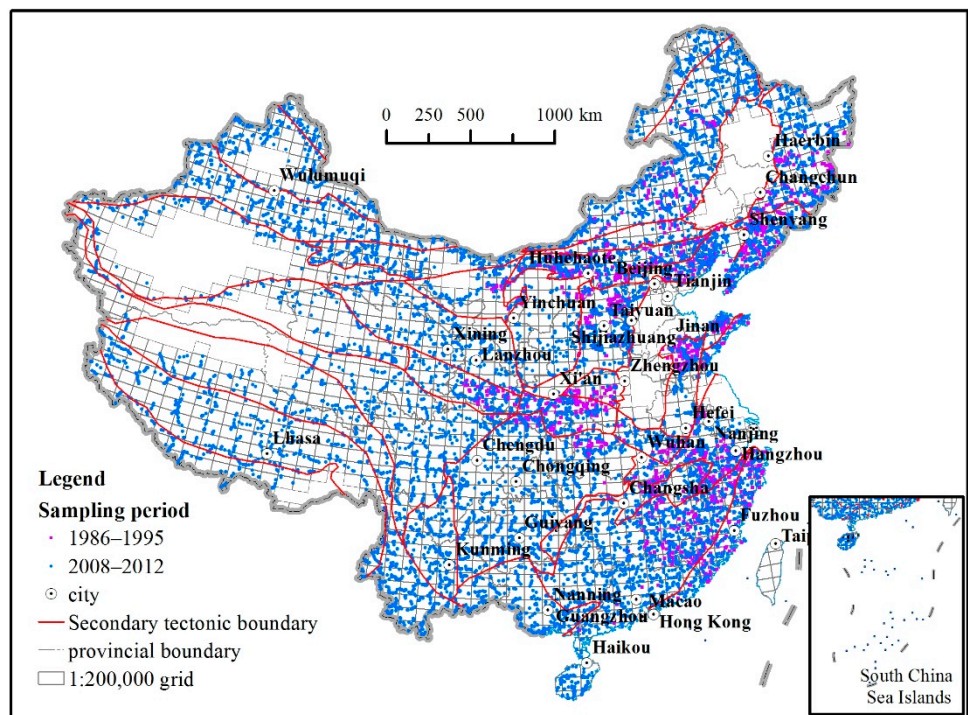

**Figure 1.** Distribution of sampling points (blue and violet) of rocks. The boundary of the secondary tectonic was drawn by Ren et al. [22].

### 2.1.2. Collection of Intrusive Rock Samples

For a small intrusive rock mass, rocks were sampled by age and lithology units, and one sample was collected from each unit. If the rock mass had clear lithofacies zoning, each lithofacies should be sampled separately and then combined into one sample. For complex rock mass and large rock mass, each lithology and facies zone should be sampled separately.

### 2.1.3. Collection of Metamorphic Rock Samples

Ortho-metamorphic rocks were sampled with reference to intrusive rocks, and para-metamorphic rocks were sampled with reference to sedimentary rocks.

### 2.1.4. Sample Preparation and Analytical Methods

Samples from both stages were analyzed in the same laboratory using the same method. The samples were crushed to less than 2 mm in diameter using a jaw crusher and then ground with an agate mortar to less than 0.075 mm in diameter. A 0.25 g sample was weighed and digested with 10 mL HF, 5 mL $HNO_3$, and 2 mL $HClO_4$ until $HClO_4$ was exhausted. The residue was dissolved with 8 mL aqua regia, then transferred to a 25 mL polyethylene test tube, and distilled water was added to a fixed volume and shaken well. One milliliter of the clarified solution was divided, diluted to 10 mL with 2% $HNO_3$, and cobalt was determined by inductively coupled plasma mass spectrometry. For the results of quality control, see Yan et al. [23] and Wang et al. [24]. The quality of the analysis met the research needs.

### 2.2. Map Producing

The production of the petrogeochemical map used a 1:200,000 geological sheet grid as the basic unit. The cobalt content of rocks is closely related to lithology, and the cobalt content of each grid can be obtained by calculating the average cobalt content of all kinds of exposed rocks in the grid by area weighting. Therefore, before mapping, it is necessary to obtain the spatial distribution information of different rock type features, and then assign cobalt content to spatial features.

### 2.2.1. Data Preparation

Spatial distribution information of different rock type features is from the spatial database of the 1:2,500,000 digital geologic map of China [25]. The rock naming system of the actual sample differed from that of the 1:2,500,000 database; the rock type was simplified to make the two correspond. The cobalt content of 44 common rock types is shown in Table 1. They can be merged into 8 groups based on the cobalt content: acid rock, intermediate-acid rock, intermediate rock, basic rock, ultrabasic rock, mudstone, sandstone, and carbonate. Figure 2 shows the cobalt content distribution of the 8 groups of rock types in this paper. It can be seen that there were significant differences in cobalt content distribution among different groups of rocks, and in the same kind of rock, the element content distribution was relatively concentrated. Element distribution characteristics indicated that the classification scheme is rational for assigning cobalt content to the exposed crust according to its rock type.

**Table 1.** Cobalt abundance of different types of rocks (mg/kg). Notation: the cobalt abundance data was quoted from Chi et al. [13].

| Ultrabasic Rocks | | Argillaceous Rocks | |
|---|---|---|---|
| **Peridotite** | **110** | **Common Mudstone** | **14.5** |
| Pyroxene peridotite | 105 | Silty mudstone | 12.4 |
| **Basic rocks** | | Calcareous mudstone | 12.5 |
| Diabase | 41 | Carbonaceous mudstone | 11 |
| Gabbro | 46 | Aluminiferous mudstone | 12 |
| Basalt | 47 | Ferroaluminiferous mudstone | 13 |
| Basic granulite | 57 | Tuffaceous mudstone | 8 |
| Amphibolite | 49 | Slates | 13 |
| Greenschist | 46 | Phyllites | 16 |
| **Intermediate rocks** | | Schist | 14 |
| Diorite | 21 | **Arenaceous rocks** | |
| Andesite | 26 | Arkose | 9 |
| Monzonite | 26 | Silty sandstone | 11 |
| Trachyandesite | 23 | Greywacke | 9.5 |
| Intermediate granulite | 28 | Calcareous Sandstone | 7 |
| Dioritic gneisses | 24 | Tuffaceous sandstone | 8.3 |
| **Intermediate-acidic rocks** | | Metasandstone | 10 |
| Granodiorite | 12 | **Carbonate** | |
| Tonalite | 15 | Limestone | 0.8 |
| Dacite | 8.7 | Argillaceous limestone | 2.9 |
| Intermediate-acidic granulite | 12 | Dolomite | 1.1 |
| Intermediate-acidic gneiss | 16 | Argillaceous dolomite | 1.6 |
| **Acidic rocks** | | | |
| Granite | 3 | | |
| Rhyolite | 2.2 | | |
| Alkali granite | 1.2 | | |
| Monzonitic granite | 4.6 | | |
| Trondhjemite | 5.3 | | |

### 2.2.2. Data Processing and Drawing of the Petrogeochemical Map of Cobalt

In Arcmap 10.8, the "intersect" tool was used to intersect with the distribution of 8 groups of rock type, secondary tectonic units, and 1:200,000 geological map grids, producing a total of 81,541 features, each of which had attributes of rock type, first-order tectonic unit, secondary tectonic units, and 1:200,000 map sheet grids. The features were assigned a cobalt content value depending on the cobalt content of the same rock type in the same 1:200,000 geological map grids, or in the same secondary tectonic unit if no data were in the same map grids. The area of each feature was calculated in Arcmap10.8, all of the

features distributed including area, cobalt content, 1:200,000 geological map grids' ID were exported as database datafile, and the cobalt content of each map sheet was calculated in Microsoft Excel using the area-weighted average method (Figure 3).

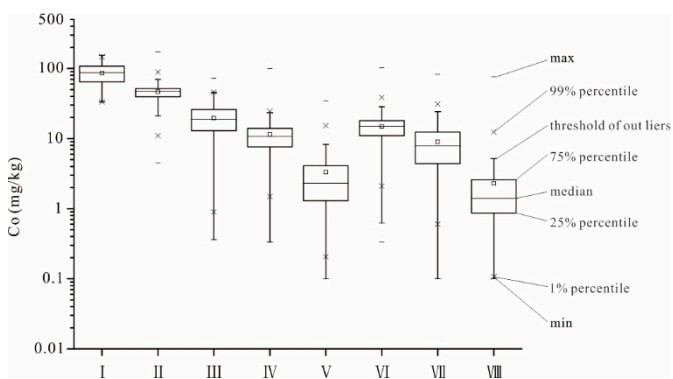

**Figure 2.** Box plot of cobalt content in different rock types in this paper. I: Ultrabasic rocks, II: Basic rocks, III: Intermediate Rocks, IV: Intermediate-acid rocks, V: Acidic rocks, VI: Mudstones, VII: Sandstones, VIII: Carbonates. Outliers are cases with values larger than 1.5 times the box length (interquartile range) from the lower or upper edge of the box.

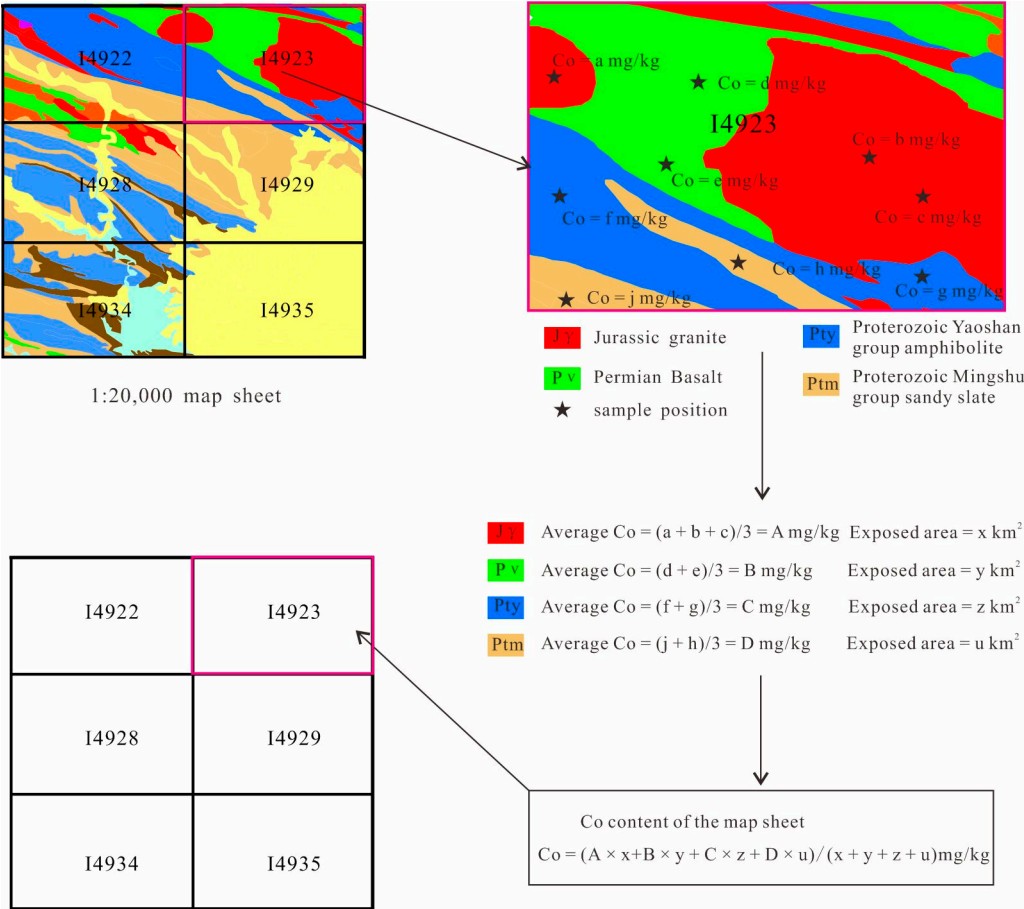

**Figure 3.** Schematic diagram of the process of assignment of cobalt content to 1:200,000 geological map grids.

The data file of the database was imported into ArcMap 10.8. The coordinate of the geometry center of each 1:200,000 geological map grid was calculated, and then the point that represented the average cobalt content of the map grid was produced. The point was

interpolated to a polygon using the inverse distance weighting (IDW) method. The power exponent was set to 2, cell size to 50 km, and search radius to 150 km.

## 3. Results and Discussion

### 3.1. Cobalt Abundance of China's Exposed Crust

There are 1421 grids calculated for the cobalt abundance of the exposed crust throughout China. Cobalt statistical parameters of the grids are shown in Table 2. The median, mean, and geomean values were 9.74 mg/kg, 10.67 mg/kg, and 9.33 mg/kg, respectively. The 50% of the grids have a cobalt abundance ranging from 7.01 mg/kg to 12.99 mg/kg. The median, mean, and geomean cobalt values of China's soil are 10.62 mg/kg, 11.42 mg/kg, 9.95 mg/kg, respectively [12]. The background cobalt value of China's soil is 10.62 mg/kg, which is slightly higher than that of the exposed crust of China. In the ground-surface environment, cobalt ions are easily adsorbed by Fe, Mn oxides, and organic carbon under specific physicochemical conditions (such as alkaline and oxidative environments), resulting in secondary cobalt enrichment in soil [26–28].

**Table 2.** Statistical parameters of cobalt abundance of exposed crust and deep soils throughout China (mg/kg). N = Number of samples; P = Percentile (P50 = median); Min. = Minimum; Max. = Maximum. The soil data were quoted from Liu et al. [12].

|  | N | Min. | P2.5 | P25 | P50 | P75 | P85 | P97.5 | Max. | Mean | Geometric Mean |
|---|---|---|---|---|---|---|---|---|---|---|---|
| Exposed crust | 1421 | 0.73 | 2.97 | 7.01 | 9.74 | 12.99 | 15.58 | 25.23 | 51.29 | 10.67 | 9.33 |
| Soil | 3380 | 0.91 | 2.92 | 7.34 | 10.62 | 14.26 | 16.30 | 25.49 | 80.79 | 11.42 | 9.95 |

Crustal abundance of cobalt was estimated between 24.00 mg/kg~29.00 mg/kg [29–31], and crustal abundance of eastern China was estimated at 19.00 mg/kg by Yan and Chi [32]. The upper crustal abundance of cobalt was estimated between 10.00 mg/kg ~18.00 mg/kg [30,31,33,34]. All above are all higher in abundance than the Chinese exposed crust given by this paper. This is because the rock composition of both the crust and the upper crust is different from that of the exposed crust. With an increase in crust depth, the proportion of basic rocks increases, and the cobalt content increases accordingly [29], so the exposed crust has a lower Cobalt abundance than the upper crust and whole crust.

Histograms and boxplots were plotted to show cobalt distribution in different grids throughout China (Figure 4). The data were close to the log-normal distribution. However, the frequencies with a "Log Co" content greater than 1.3 mg/kg (the equivalent cobalt content is 19.5 mg/kg) decreased sharply. Cobalt content in rocks is closely related to rock type. The cobalt content of 19.5 mg/kg mainly corresponds to intermediate rocks, basic rocks, and ultrabasic rocks (Figure 2). According to the geological map of China [25], their exposed areas are much smaller than those of clastic rock, acid igneous rock, and carbonate rock, which are not rich in cobalt. Thus, the histogram reflects the distribution of cobalt in the exposed curst.

### 3.2. Spatial Variation of Cobalt Abundance in the Exposed Crust

The spatial distribution of cobalt in the exposed crust throughout China is shown on the contoured petrogeochemical map (Figure 5). The empty areas in Xinjiang, Henan-Hebei, and Northeast China are covered by thick Quaternary sediments. The different color shades given to percentiles between 0~2.5 (dark blue), 2.5~25 (blue), 25~50 (light blue), 50~75 (green), 75~85 (yellow), 85~97.5 (red), and 97.5~100 (dark red) were selected to facilitate the reading and interpretation of Cobalt.

Low Co concentrations (<P25, blue colors) are mainly distributed in southeast China, northeast China, and Tibet, where large areas of acid rock and carbonate are exposed (Figure 6). Take the largest area with a low Co concentration located on the southeast coast and Jiangxi-Guangxi-southeastern Guizhou as an example. The rocks are mainly Cretaceous rhyolite volcanic rocks, contemporaneous aluminum granite rocks in the southeast coastal

area, and Cambrian, Upper Paleozoic, and Triassic carbonate strata, a small amount of Jurassic granite in Jiangxi-Guangxi-southeastern Guizhou [35].

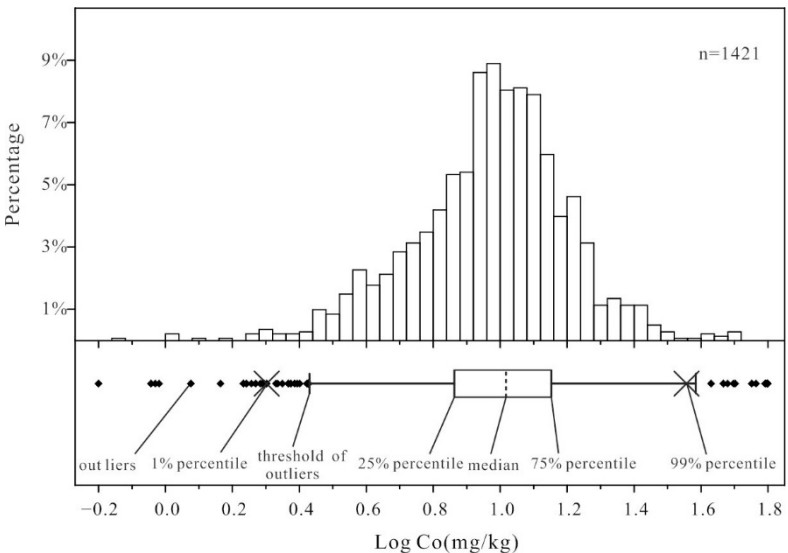

**Figure 4.** Histogram and boxplot displaying cobalt distribution in different grids throughout China. Notation: Outliers are cases with values larger than 1.5 times the box length (interquartile range) from the lower or upper edge of the box.

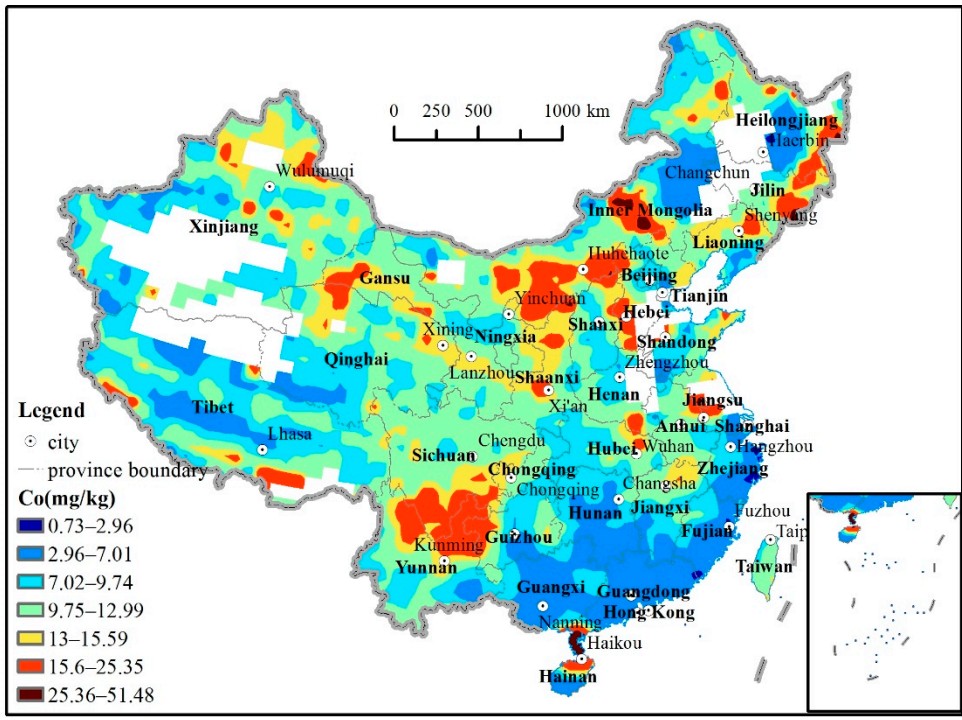

**Figure 5.** Petrogeochemical map of cobalt of exposed crust in China.

High Co concentrations (P75~P85, yellow colors) are mainly distributed in the junction of Yunnan, Guizhou, and Chongqing in southwest China, and northwest China, including Inner Mongolia, Shaanxi, Gansu, and northern Xinjiang. There is also scattered distribution in north and east China. The rocks exposed in the above area are mainly intermediate rocks and argillaceous rocks [25].

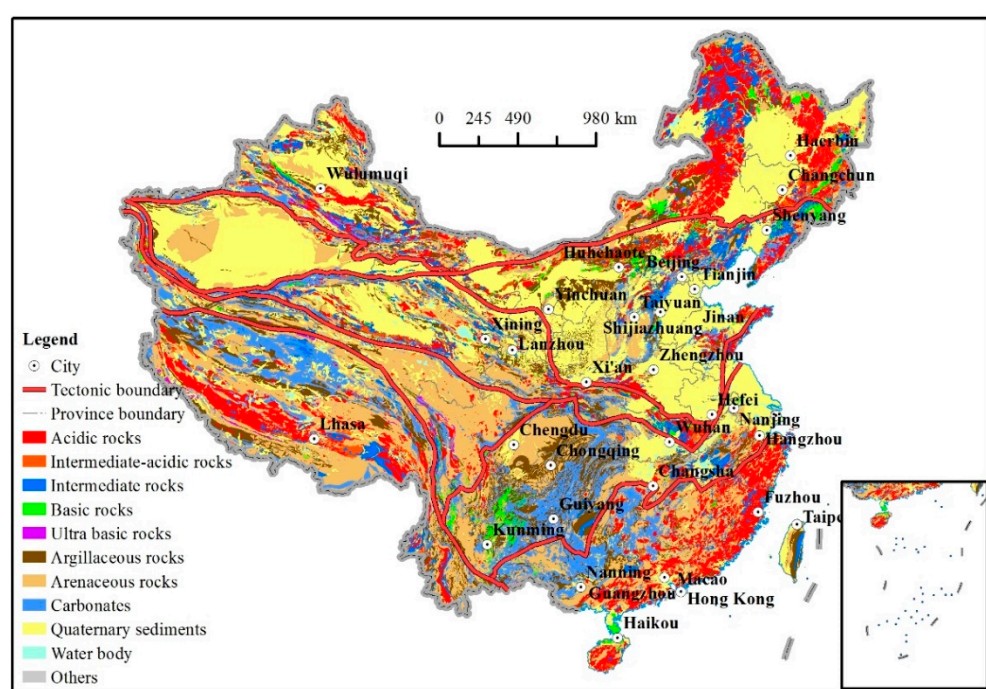

**Figure 6.** Lithology distribution and tectonic boundary map of China (lithology distribution based on Ye et al. [25], tectonic boundary redraw from Ren et al. [22]).

Co anomalies (Threshold here is defined as >P85, red colors) are distributed in southwest, north, and northwest China. The southwest anomaly is related to the basalts of Permian Emeishan Large Igneous Province (ELIP) and upper Permian sedimentary rocks that were formed by sediment volume backfilling during the tectonic uplift of the central ELIP [36]. The northeast anomaly is related to the Changbaishan Cenozoic basalt. The north anomalies are related to Archean metamorphic basic rocks and Cenozoic Hannuoba Formation basalts. Some small anomalies in Tibet, Xinjiang, and Gansu are related to ophiolites or ophiolite belt, such as Neo-Tethyan Cenozoic ophiolites, Tianshan late Paleozoic ophiolites, and Qinling–Qilian–Kunlun early Paleozoic ophiolite belt [37].

Cobalt anomalous centers (>P97.5, dark red) are located in central Inner Mongolia, southern Jilin, and southern Guangdong, where a large area of basalts is exposed, including the Abaga Formation of the Upper Pleistocene, Junjianshan Formation of the Lower Pleistocene, and Shimaoling Formation of the Mid-Pleistocene. The formation of the Cenozoic basalts described above is related to the subduction of the Pacific plate and the big mantle wedge (BMW); the basalts were interpreted as a mixture of high-Si melts and low-Si melts. Wherever they are from northeast, north, or south China, Cenozoic basalts share a common low-Si basalt endmember, which is characterized by high alkali, Fe, and Ti contents [38].

### 3.3. Cobalt Distribution in Relation to China's Tectonic Framework

China can be divided into eight first-order geotectonic units, which are Altay–Mongolia–Hinggan Orogens, North China Craton, Qin–Qi–Kun Orogens, Yangtze Craton, Cathaysia Foldbelt, Tarim Craton; Songpan–Ganzi Terrane and Tibet–Sanjiang Orogens [22] (Figure 7). During the differentiation of a basaltic magma, most of the cobalt enters ferromagnesian minerals. Cobalt is strongly coherent with magnesium in granitic rocks and behaves like magnesium in its partition relations between metamorphic minerals. In sediments, most of the cobalt is in the argillaceous fraction and seems to follow iron and manganese [14]. Therefore, cobalt tends to be enriched in structural units where basic rocks are widely exposed, such as craton and suture.

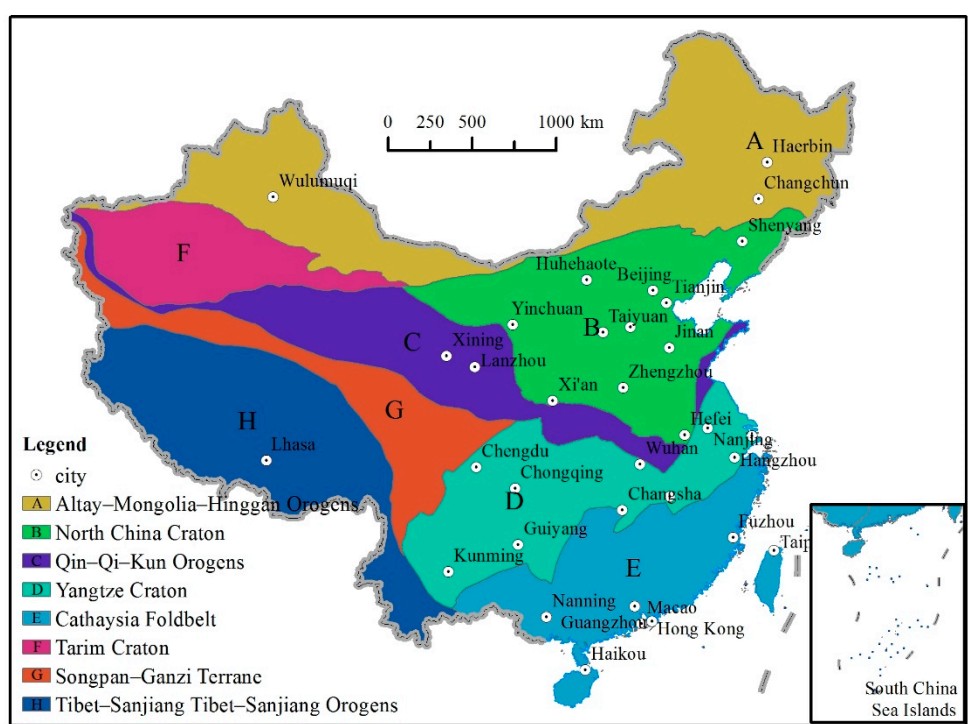

**Figure 7.** Distribution map of first-order tectonic units in China (redraw from Ren et al. [22]).

Altay–Mongolia–Hinggan Orogens are a huge system of orogenic collages, also named the Altaids, or the Central Asian Orogens [39,40], experienced a long period of accretionary-type arc-continent collision orogeneses, which started at about 1.0 Ga and lasted until about 250 Ma, at which time the Paleo-Asiatic ocean closed and formed the Tianshan–Solonker suture zones [41]. In the post-collision stage, reworking of the juvenile oceanic crust resulted in felsic magmatism and associated mineralization [42]. The North China Craton (NCC) is one of the world's oldest cratons. It is traditionally considered to be composed of Mesoproterozoic to Cenozoic sedimentary covers overlying homogeneous basements from Archean to Paleoproterozoic. However, in recent 20 years, new data have made it widely believed that NCC rocks were formed by the accretion and amalgamation of micro-continental blocks [43,44]. The South China block can be divided into the Yangtze craton and the Cathaysia Foldbelt, which are believed to have collided along the Jiangnan orogens in the Neoproterozoic. The Yangtze craton is composed of Archean—Proterozoic crystalline basement, which is unconformably covered by weakly metamorphic Neoproterozoic strata and unmetamorphosed Sinian covers. The Cathaysia Foldbelt is mainly composed of Neoproterozoic basement rocks, a few Paleoproterozoic basement rocks are found in southwest Zhejiang and northern Fujian, and Mesoproterozoic basement rocks are found in Hainan Island [44,45]. The South China Block, composed of Yangtze Craton and Cathaysia Foldbelt, has experienced long-term tectonic evolution with at least four region-scale geodynamic events [46]. Qin–Qi–Kun Orogens have two sub-orogens; the eastern part is Qinling–Tongbai–Hong'an–Dabie–Sulu orogen. The western part is the Qilian–Kunlun orogen. The Qinling-Tongbai–Hong'an–Dabie–Sulu orogen formed during the collision of Paleozoic and Mesozoic continental blocks, forming ultra-high pressure metamorphic rocks [44]. The Qilian–Kunlun orogen is mainly composed of amphibolite facies gneiss, schist, and migmatite and is overlain by sedimentary sequences that are partially intruded by gneissic potassic granite [47]. Tarim craton is located in western China, where Neoarchean and Paleoproterozoic crystalline basements are developed. It is overlaid by late Mesoproterozoic to mid-early Neoproterozoic metamorphic greenschist and blueschist facies deposits and volcanic strata, which are overlaid by late Neoproterozoic Nanhua and Sinian unmetamorphosed covers [32]. Tarim Basin is the largest sedimentary basin in China, with an area of about 560,000 square kilometers and a sedimentary thickness

of 5–20 km [44]. It began to become an inland basin in the Mesozoic era and is now covered by the Taklimakan desert, with Paleozoic rocks distributed mainly on its northern and southern margins. Although Paleozoic magmatism is not obvious in the Tarim Basin, the large igneous province of Early Permian widely existed in the Tarim block, which is a kind of continental flood basalt [48]. The Songpan–Ganzi Terrane (also called Songpan–Ganzi Orogen) is located at the intersection of the Tethyan orogenic domain and the Paleo-Asian Ocean orogenic domain, and it is a tectonic unit that has continuously evolved since the Late Paleozoic [49]. It comprises >200,000 km$^2$ of 5–15 km thick turbiditic sediments. The Qinling–Tongbai–Hong'an–Dabie orogen, located several hundred kilometers to the east, is its major source [50]. Tibet–Sanjiang Orogens (also known as the Tibetan Plateau) are arc-continental collisional orogens consisting of a few microcontinents, such as Qiangtang and Lhasa terranes. During the Late Paleozoic to Early Mesozoic, these microcontinents moved northward, colliding with the southern active margin of the Eurasian continent and forming multiple sutures in and out of the Tibet Plateau [51,52]. From Cretaceous to Eocene, the northward moving of the Indian continent collided with the Eurasian continent [53], as a result, Cenozoic granitic rocks are more common in the Himalayan orogenic belt and southern Tibet block [54].

Table 3 shows the cobalt abundance of the different tectonic units, and Figure 8 shows the distribution characteristics of the cobalt content in different tectonic units. The median value in ascending order is: Cathaysia Foldbelt < Tibet–Sanjiang Orogens < Altay–Mongolia–Hinggan Orogens < whole China < Tarim Craton < Songpan–Ganzi Terrane < Yangtze Craton < Qin–Qi–Kun Orogens < Cratons < North China Craton. Except for Songpan–Ganzi terranes and Qin–Qi–Kun Orogens, the cobalt abundance of all cratons was higher than that of orogens. It is mainly related to the uplift emplacement of granites induced by orogenic processes [55,56]. Among the five orogens, the cobalt abundance of the Precambirian Cathaysia Foldbelt is the lowest, and that of the Phanerozoic Altay–Mongolia–Hinggan Orogens is the highest. Unlike the crust-mantle mixed source areas of granites in other orogenic belts, the source areas of granites in Cathaysia Foldbelt are mainly crust-cycled and therefore have lower cobalt abundance [57]. However, the granites in the Altay–Mongolia–Hinggan Orogens are mainly of mantle origin, and almost no crust-derived materials are added [58]. In addition, several orogenic belts are relatively rich in cobalt, which is related to several Paleozoic ophiolite belts distributed in them, such as the Qinling–Qilian–Kunlun ophiolite belt, West Junggar ophiolites, East Junggar ophiolites, Tianshan Ophiolites, and Inner Mongolian Ophiolite, which are conducive to the elevation of cobalt abundance.

**Table 3.** Statistical parameters of cobalt abundance (mg/kg) of 1421 grids in different tectonic units.

| | Sample Quantity | Min | 2.50% | 25% | 50% | 75% | 85% | 98% | Max | Mean | Geometric Mean |
|---|---|---|---|---|---|---|---|---|---|---|---|
| **Whole China** | 1421 | 0.7 | 3 | 7 | 9.7 | 13 | 15.6 | 25.2 | 51.3 | 10.7 | 9.3 |
| **Orogens** | 982 | 1 | 2.8 | 6.3 | 9.3 | 12.3 | 14.3 | 23.4 | 50.7 | 10 | 8.7 |
| Altay–Mongolia–Hinggan Orogens | 375 | 1.6 | 2.9 | 7.2 | 9.6 | 13.2 | 15.6 | 25.8 | 41.8 | 10.7 | 9.3 |
| Qin–Qi–Kun Orogens | 150 | 3.7 | 5.6 | 8.8 | 11.1 | 14.3 | 15.8 | 18.9 | 21.9 | 11.6 | 11 |
| Cathaysia Foldbelt | 159 | 1.3 | 2.3 | 4 | 5.5 | 7.9 | 9.9 | 23.7 | 50.7 | 7.4 | 6 |
| Songpan–Ganzi Terrane | 88 | 3.4 | 3.8 | 7.6 | 10 | 11.8 | 13.4 | 19.4 | 27.1 | 10.3 | 9.5 |
| Tibet–Sanjiang Orogens | 210 | 1 | 2.8 | 6.4 | 9.1 | 11.1 | 12.4 | 23 | 29.7 | 9.5 | 8.4 |
| **Cratons** | 439 | 0.7 | 3.9 | 8.2 | 11.2 | 14.6 | 17.2 | 26.4 | 51.3 | 12.2 | 10.9 |
| North China Craton | 230 | 0.7 | 3.7 | 8.5 | 12 | 15.8 | 17.5 | 29.6 | 51.3 | 12.9 | 11.3 |
| Yangtze Craton | 151 | 2.8 | 4.2 | 7.9 | 10.8 | 14 | 17.2 | 25.1 | 32.5 | 11.9 | 10.7 |
| Tarim Craton | 58 | 3.5 | 4.4 | 7.5 | 9.7 | 11.5 | 12.3 | 19.9 | 25.9 | 10.1 | 9.5 |

Although the North China Craton and the Yangtze Craton are spatially adjacent, they have independent geochemical evolution systems [59]. Unlike the Yangtze Craton and Tarim Craton, where sedimentary covers are well developed, the North China Craton has a complex history of collage and disruption, with extensive Archean intermediate-basic

metamorphic rocks and Mesozoic-Cenozoic basalts [38,60–62]. The reason above leads to higher cobalt abundance values in the North China Craton.

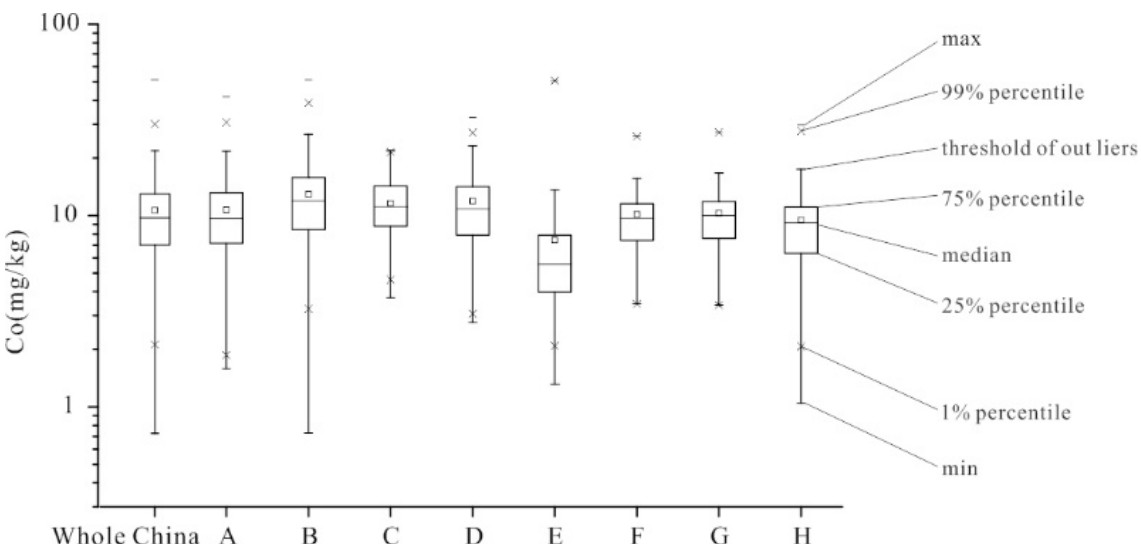

**Figure 8.** Boxplots showing the Co (µg/kg) abundance of 1:200,000 map sheet grids in different tectonic units of China (for summary statistics, refer to Table 3). Notation: (A) Altay–Mongolia–Hinggan Orogens; (B) North China Craton; (C) Qin–Qi–Kun Orogens; (D) Yangtze Craton; (E) Cathaysia Foldbelt; (F) Tarim Craton; (G) Songpan–Ganzi terrane; (H) Tibet–Sanjiang Orogens.

### 3.4. Verification of Cobalt Abundance of Exposed Crust

The biggest challenge in petrogeochemical mapping is the drastic change in element content between different geological bodies. This problem may not be solved by increasing sampling densities only, so we chose to assign values to similar geologic bodies. This method was verified by spatial correlations between the cobalt content of sediments and rocks. Although the secondary enrichment or depletion of chemical elements occurs during the transformation of rocks into stream sediments, the cobalt content in rocks and sediments should be well correlated if they are in the same supergene environment or landscape. Therefore, the correlation between the cobalt content of sediments and rocks can be used to estimate the accuracy of the calculated cobalt abundance in the exposed crust.

The Regional Geochemistry-National Reconnaissance (RGNR) Project of China was carried out from 1978 to the early 21st century with the aim of mineral exploration [63]. Stream sediment samples were collected at a density of 1 to 2 samples per square kilometer, and the cobalt was determined by XRF with a detection limit of 0.1 mg/kg [64]. A large number of theoretical studies and exploration achievements have fully demonstrated that the RGNR project has a high quality of sampling and analysis [65]. The overlap area between RGNR and petrogeochemical mapping in this paper is shown as grids in Figure 9. China has a vast territory, and different regions have great differences in topography, climate, and temperature, forming 14 typical landscapes (Figure 9) [66]. There were obvious differences in the migration and enrichment behaviors of chemical elements in different landscape areas. Therefore, the correlation study needs to take into account changes in the landscape. The correlation study of the two sets of data involved 11 of the 14 landscapes, not involving desert, hilly grassland, and alluvial plain, due to the small number of samples in those landscapes having no statistical significance.

Each of the 1051 grids was assigned a mean cobalt content of RGNR stream sediment samples and cobalt abundance of exposed crust. The Pearson correlation coefficient was used to evaluate the cobalt content correlation between RGNR and exposed crust for different landscapes and to verify the reliability of the cobalt abundance in the exposed crust given in this paper (Table 4).

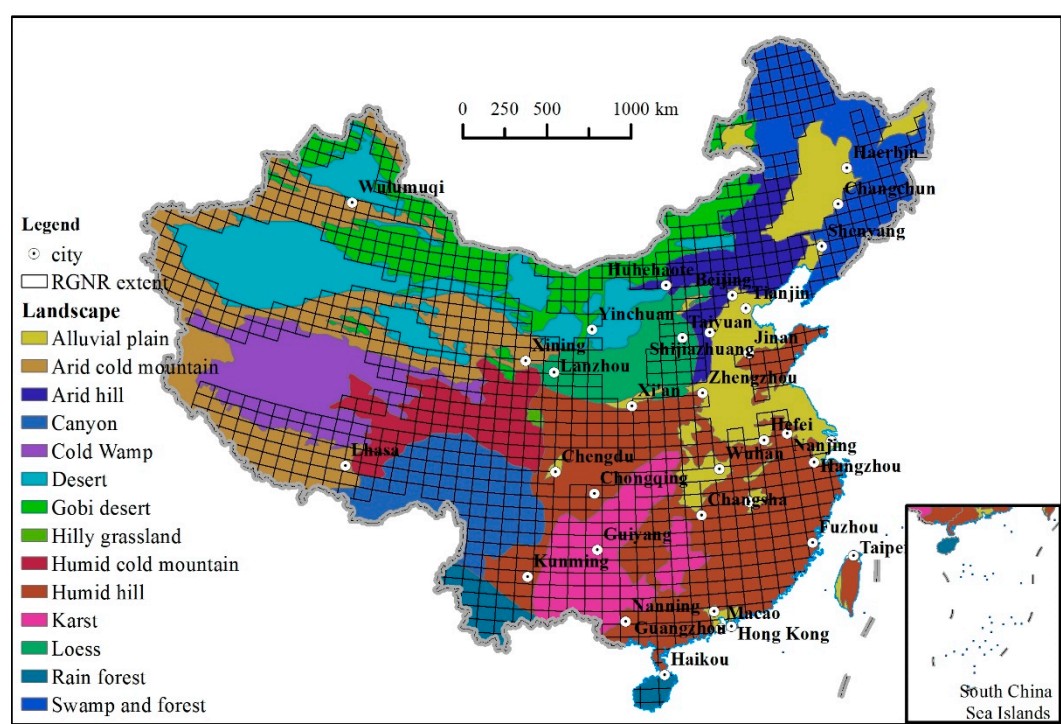

**Figure 9.** Map of RGNR extent and landscapes across China (Distribution of landscape was redrawn by Guo et al. [66]).

**Table 4.** Grid-based data correlation and statistical characteristics of cobalt. The data of stream sediments was obtained from Xie et al. [64].

| Landscape | Number of Grids | Coefficient of Correlation | Co of Exposed Crust (mg/kg) | Co of Stream Sediments (mg/kg) | Enrichment Coefficient |
|---|---|---|---|---|---|
| All | 1051 | 0.31 | 10.49 | 12.08 | 1.15 |
| Humid hill | 233 | 0.66 | 9.87 | 13.00 | 1.32 |
| Karst | 67 | 0.61 | 8.28 | 20.03 | 2.42 |
| Tropical rainforest | 34 | 0.59 | 9.10 | 12.41 | 1.36 |
| Canyon | 65 | 0.57 | 12.53 | 15.74 | 1.26 |
| Arid Hill | 58 | 0.55 | 11.75 | 10.99 | 0.94 |
| Humid cold mountain | 57 | 0.41 | 9.69 | 11.37 | 1.17 |
| Swamp and forest | 127 | 0.34 | 11.44 | 12.91 | 1.13 |
| Cold swamp | 39 | 0.30 | 8.50 | 8.11 | 0.95 |
| Loess | 37 | 0.28 | 11.53 | 11.45 | 0.99 |
| Arid cold mountain | 197 | 0.26 | 10.01 | 10.29 | 1.03 |
| Gobi Desert | 137 | 0.14 | 11.40 | 9.15 | 0.80 |

The correlation coefficients of different landscape areas are very different, and they are 0.55~0.66 in mountainous regions with well-developed stream systems, including Humid hill, Karst, Tropical rainforest, Canyon, and Arid Hill. The above result indicates a good relationship between the cobalt content of RGNR stream sediment samples and the cobalt abundance of the exposed crust. However, in landscapes such as Cold swamp, Loess, Arid cold mountain, and Gobi Desert with less developed stream systems, the correlation coefficients are lower than 0.34, which represents a low degree of correlation. Through the above verification results, it can be concluded that the results of the rock geochemical mapping carried out in this paper are reliable.

## 4. Conclusions

Petrogeochemical mapping of cobalt covering the whole of China has been carried out for the first time. The spatial variation at the surface of cobalt abundance was plotted based on a 1:200,000 geological map grid using more than 38,000 rock samples combined with the spatial distribution information database of exposed lithology. The abundance of cobalt in the exposed crust can provide the geological background of cobalt for geochemical exploration and environmental evaluation.

The cobalt abundance of the exposed crust in China is 9.74 mg/kg, which is lower than the crust abundance and background of soils. The abundances are related to the vertical differentiation of cobalt in the crust and the secondary enrichment of cobalt in the epigenetic environment. Co anomalies and anomalous centers of exposed crust are distributed in southwest, north, and northwest China, where they are primarily related to Permian Emeishan Large Igneous Province (ELIP), Pacific plate, and the big mantle wedge (BMW), Archean metamorphic basic rocks, and ophiolite belts.

**Author Contributions:** Conceptualization, L.N., F.Y., Q.C. and X.W.; methodology, D.L.; software, Y.C.; validation, D.L.; formal analysis, D.L.; investigation, D.L.; resources, Q.C. and X.W.; data curation, Y.C.; writing—original draft preparation, D.L.; writing—review and editing, F.Y. and L.N.; visualization, Y.C., L.N. and F.Y.; project administration, Q.C. and X.W. All authors have read and agreed to the published version of the manuscript.

**Funding:** This research was funded by 'Sino-probe 04-01 (Grant no. 201011053)', 'Basic Geological project (Grant no. 8502212)', 'the National Nonprofit Institute Research Grant of IGGE (Grant no. AS2020Y06 and Grant no. AS2020J06)' and 'the National Resources Survey Projects (Grant no. DD20221641)'.

**Acknowledgments:** Many thanks are given to all participants in the 'China Geochemical Baselines' and 'Element abundance in the upper crust of eastern China'.

**Conflicts of Interest:** The authors declare no conflict of interest.

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
