# Peer review of "National-Scale Cobalt Geochemical Mapping of Exposed Crust in China"

_minerals, doi:10.3390/min12101220_

Round 1

Reviewer 1 Report

The manuscript presents the mapping of the cobalt content in the P.R. of China. The idea presented is very interesting, and the manuscript can be accepted with some minor revisions. 

1. Line 71-78. This part of the manuscript describes the reasons why the research was conducted. I think the authors should extend this part according to the toxicity of the Cobalt in the environment. This last issue can improve the novelty of the entire research. Also, considering that Cobalt is on the list of critical materials, it would be nice to understand if the authors can also indicate where Cobalt is found in association with the ores used for cobalt processing. 

2. We have a huge number of samples (38,899) collected in two different time frames. What is not clear is if the analytical analyses were determined for each of the two sampling time frames or during a third time frame. It is also helpful to understand whether the ICP-MS used is only one or if the samples where determined with the support of different ICP-MSs. In this last case, it is essential to know if the limits of detection, accuracies, and others are the same for each instrument.   

3. Citations shall follow the journal's policies. Please, correct them in all the manuscript. 

4. There are some empty spaces on the maps (Xinjiang, Henan-Hubei, Dongbei, and so on). What is the reason? Lack of official maps? Please, explain it better in the text. 

5. In paragraph n. 3, the authors discuss the details of the distribution of the Cobalt. It would be nice if the authors could add brief descriptions of the general behavior of Cobalt in different petrogenetic environments. 

Best Regards

Author Response

Dear Editor and Reviewers: On behalf of my co-authors, we are grateful to you for giving us an opportunity to revise our manuscript. We appreciate you very much for your valuable comments. We have studied both reviewers’ comments. The following are the responses and revisions to reviewer #1 on an item-by-item basis.

  1. Line78-82, extended the reasons why this research was conducted: “The Democratic Republic of the Congo, South Africa, Australia, Philippines, etc are major cobalt ores exporters [17], and China has been a leading consumer and refiner of cobalt[18]. For most countries including China, the recycling efficiency of cobalt is rel-atively low, which means large amounts of cobalt are released into the environment[17]. The pollution of cobalt as heavy metal has also been a concern in re-cent years[19,20].”
  2. Line 107, added ‘Samples from both stages were analyzed in the same laboratory using the same method.’
  3. Corrected the citations according to the journal's policies.
  4. Line 203-204, added ‘The empty area in Xinjiang, Henan-Hebei and Northeast China are covered by thick Quaternary sediments.’ to explain why these regions lack of rocks data.
  5. Line 247-252, added ‘During differentiation of a basaltic magma, most of the cobalt enters the ferromagne-sian minerals. Cobalt is strongly coherent with magnesium in granitic rocks and be-haves like magnesium in its partition relations between metamorphic minerals. In sediments most of the cobalt is in the argillaceous fraction and seems to follow iron and manganese[13]. Therefore, cobalt tends to be enriched in structural units where basic rocks are widely exposed such as craton and suture.’

Reviewer 2 Report

Overview:

The authors collected more than 38 000 samples of exposed crust (outcrops) in all of China and analyzed them for cobalt. The data were used to create a geochemical map. The results showed a median cobalt concentration of 9.74 mg/kg. Areas of higher concentration were noted in  southwest and northwest China whereas areas with lower concentrations were found in the east. The authors propose that this distribution is related to China’s tectonic framework.

Overall, the paper is well written and the methodologies employed are sound. All figures are appropriate. I think the results and approach will be of interest to readers elsewhere and I support that the paper is published in Minerals after addressing the minor revisions listed below.

Comments:

1. Line 38, change “Secondly, the mineralization..”  to “Secondly, mineralization…”

2.     Line 41, replace ‘clearly’ with ‘entirely’

3.     Line 52, change ‘earth’ to ‘Earth’

4.     Line 65, change ‘km2’ to ‘km2

5.     Line 67, replace ‘even’ with ‘anywhere’

6.     Line 199, replace ‘make easier’ with ‘facilitate’

7.     Line 237, ‘Altay-Mongolia-Hinggan-Orogens’ are not shown in the legend in Figure 7. Is a different name used? Please have this corrected to make it easier for the reader to follow your discussion.

8.     Line 237, similarly, the Cathaysta Foldbelt is not shown in Figure 7. Please be consistent in your labeling or provide additional explanation for where this should be.

9.     Line 243, Orogens are plural. Also, first letter in ‘orogens’ is capitalized elsewhere in the paper. So, change “Altay-Mongolia-Hinggan orogens is..” to “Altay-Mongolia-Hinggan Orogens are..”

10.  Line 248, change “The North China Craton..” to “The North China Craton (NCC)..”  so that you can use the abbreviation in line 252. As currently written the abbreviation is used without having been defined on first use.

11.  Line 331, the year 1978 was already in the 20th century which started in 1901. So, I think you should change “..from 1978 to the early 20th century….”  to “..from 1978 to the early 21st century….”  

12.  Line 370, what is ‘they’ in “…background of soils, they are related..” If this refers to abundances then this should be stated to avoid confusion. I suggest changing it to “…background of soils. The abundances are related..”

Author Response

On behalf of my co-authors, we are grateful to you for giving us an opportunity to revise our manuscript. We appreciate you very much for your valuable comments. We have studied both reviewers’ comments. The following are the responses and revisions to reviewer #2 on an item-by-item basis.

  1. Line 38, changed “Secondly, the mineralization..”to “Secondly, mineralization…”
  2. Line 41, replaced ‘clearly’ with ‘entirely’
  3. Line 52, changed ‘earth’ to ‘Earth’
  4. Line 65, changed ‘km2’ to ‘km2
  5. Line 67, replaced ‘even’ with ‘anywhere’
  6. Line 206, replaced ‘make easier’ with ‘facilitate’
  7. In the legend of Figure 7., ‘Xingan-Mongolia Orogenic belt’ was changed to ‘Altay-Mongolia-Hinggan-Orogens’.
  8. In the legend of Figure 7., ‘South China Orogenic belt’ was changed to ‘Cathaysta Foldbelt’.
  9. Line 255, changed “Altay-Mongolia-Hinggan orogens is..” to “Altay-Mongolia-Hinggan Orogens are..”

10.Line 261, changed “The North China Craton..” to “The North China Craton (NCC)..”.

11.Line 343, changed “..from 1978 to the early 20th century….”to “..from 1978 to the early 21st century….”.

12.Line 382, ‘they’ refers to abundances, so changed “…background of soils, they are related..” to “…background of soils. The abundances are related..”